# Pre-Operative Decitabine in Colon Cancer Patients: Analyses on WNT Target Methylation and Expression

**DOI:** 10.3390/cancers13102357

**Published:** 2021-05-13

**Authors:** Janneke F. Linnekamp, Raju Kandimalla, Evelyn Fessler, Joan H. de Jong, Hans M. Rodermond, Gregor G. W. van Bochove, Frans O. The, Cornelis J. A. Punt, Willem A. Bemelman, Anthony W. H. van de Ven, Pieter J. Tanis, Elles M. Kemper, Lianne Koens, Evelien Dekker, Louis Vermeulen, Hanneke W. M. van Laarhoven, Jan Paul Medema

**Affiliations:** 1Laboratory for Experimental Oncology and Radiobiology (LEXOR), Center for Experimental and Molecular Medicine, Amsterdam UMC, University of Amsterdam, Cancer Center Amsterdam, Meibergdreef 9, 1105 AZ Amsterdam, The Netherlands; j.f.linnekamp@amsterdamumc.nl (J.F.L.); rajbiochem@gmail.com (R.K.); fessler@genzentrum.lmu.de (E.F.); j.h.dejong@amsterdamumc.nl (J.H.d.J.); h.rodermond@amsterdamumc.nl (H.M.R.); g.g.vanbochove@amsterdamumc.nl (G.G.W.v.B.); l.vermeulen@amsterdamumc.nl (L.V.); 2Oncode Institute, 1105 AZ Amsterdam, The Netherlands; 3Department of Gastroenterology and Hepatology, OLVG, 1105 AZ Amsterdam, The Netherlands; fransolivier.the@usz.ch; 4Department of Medical Oncology, Cancer Center Amsterdam, Amsterdam UMC, University of Amsterdam, 1105 AZ Amsterdam, The Netherlands; c.j.a.punt@umcutrecht.nl (C.J.A.P.); h.vanlaarhoven@amsterdamumc.nl (H.W.M.v.L.); 5Department of Surgery, Amsterdam UMC, University of Amsterdam, Cancer Center Amsterdam, 1105 AZ Amsterdam, The Netherlands; w.a.bemelman@amsterdamumc.nl (W.A.B.); a.w.vandeven@amsterdamumc.nl (A.W.H.v.d.V.); p.j.tanis@amsterdamumc.nl (P.J.T.); 6Department of Surgery, Flevo Hospital Almere, 1315 RA Almere, The Netherlands; 7Department of Pharmacology, Amsterdam UMC, University of Amsterdam, 1105 AZ Amsterdam, The Netherlands; e.m.kemper@amsterdamumc.nl; 8Department of Pathology, Amsterdam UMC, University of Amsterdam, 1105 AZ Amsterdam, The Netherlands; l.koens@amsterdamumc.nl; 9Department of Gastroenterology and Hepatology, Amsterdam UMC, University of Amsterdam, Cancer Center Amsterdam, 1105 AZ Amsterdam, The Netherlands; e.dekker@amsterdamumc.nl

**Keywords:** decitabine, colon cancer, DNA methylation, clinical translation study

## Abstract

**Simple Summary:**

Colon cancer is one of the leading causes of cancer-related death worldwide. Therefore, the development of new therapeutic strategies is of the utmost importance. Previously, we identified a subset of colon cancers that are characterised by DNA methylation and have a poor prognosis. In this study, we therefore treated ten colon cancer patients with a demethylating agent, decitabine, to investigate if reversal of methylation is feasible and can be used as a novel therapy. Unfortunately, this study revealed that while decitabine treatment is effective in vitro, it only marginally decreased global methylation in patients and had no effect on the specific regions of DNA methylation in the tumours. Future studies should therefore focus on optimisation of treatment schedules in patients with highly methylated tumours.

**Abstract:**

DNA hypermethylation is common in colon cancer. Previously, we have shown that methylation of WNT target genes predicts poor prognosis in stage II colon cancer. The primary objective of this study was to assess whether pre-operative treatment with decitabine can decrease methylation and increase the expression of WNT target genes *APCDD1*, *AXIN2* and *DKK1* in colon cancer patients. A clinical study was conducted, investigating these potential effects of decitabine in colon cancer patients (DECO). Patients were treated two times with 25 mg/m^2^ decitabine before surgery. Methylation and expression of *LINE1* and WNT target genes (primary outcome) and expression of endogenous retroviral genes (secondary outcome) were analysed in pre- and post-treatment tumour samples using pyrosequencing and rt-PCR. Ten patients were treated with decitabine and eighteen patients were used as controls. Decitabine treatment only marginally decreased *LINE1* methylation. More importantly, no differences in methylation or expression of WNT target or endogenous retroviral genes were observed. Due to the lack of an effect on primary and secondary outcomes, the study was prematurely closed. In conclusion, pre-operative treatment with decitabine is safe, but with the current dosing, the primary objective, increased WNT target gene expression, cannot be achieved.

## 1. Introduction

The genetic aberrations in colon cancer have been extensively studied and are traditionally described by “the Vogelgram”, starting with loss of functional *APC*, followed by mutations in other genes including *KRAS*, *TP53* and *SMAD4* [1]. In addition to genetic events, epigenetic alterations are frequently found and have been shown to be essential for the initiation and progression of colon cancer [2]. DNA methylation is associated with changes in the chromatin structure and results in altered gene expression without permanently changing the DNA sequence itself [3]. In various types of tumours, genome-wide hypomethylation mainly occurs in repetitive sequences and can lead to genomic instability [4,5]. In contrast, DNA hypermethylation occurs in CpG islands in promotor regions of specific genes, resulting in transcriptional silencing (e.g., tumour suppressor genes), methylation of *CDKN2A* in many cancers being an example [3,6,7]. Besides being an important step in tumourigenesis, DNA hypermethylation has also been suggested to cause resistance to systemic therapy [8,9].

In colon cancer, relevant tumour suppressor genes are epigenetically silenced by DNA hypermethylation. For example, silencing of *MLH1*, a DNA mismatch repair gene, results in microsatellite instable (MSI) tumours. Methylation of *MLH1* as well as other genes is encompassed in the CpG island Methylator Phenotype (CIMP). This phenotype is characterised by global hypermethylation and, in proximal tumours, is associated with worse prognosis [10,11]. Methylation of several other genes with biological, predictive or prognostic relevance has also been reported [12,13]. Previously, we have shown that methylation of the WNT target genes *APCDD1*, *AXIN2* and *DKK1* predicts poor prognosis in stage II colon cancer patients [14,15]. These genes can be methylated in both CIMP high, low and negative samples and are all negative regulators of the WNT pathway by negative feedback [16,17,18]. Therefore, inactivation of these genes by methylation can lead to activation of the WNT pathway. Importantly, even in *APC* mutant CRC, some level of WNT pathway regulation is still observed and inactivation of WNT pathway inhibitors is therefore thought to further tune the pathway.

DNA hypermethylation is facilitated by a group of enzymes called DNA methyltransferases (DNMTs) [19]. Azacitidine and decitabine are the best-known examples of DNMT inhibitors and are FDA-approved for myelodysplastic syndrome (MDS) and acute myeloid leukaemia [20,21]. In preclinical studies, we showed re-expression of WNT target genes in xenografted tumours after treatment of mice with the demethylating agent azacitidine [14]. Moreover, a subsequent decrease in tumour growth was observed [14]. These findings suggest that DNA methylation could be a therapeutic target in colon cancer and inducing re-expression could potentially lead to improved patient outcomes, especially in tumours characterised by extensive WNT target gene methylation.

The relationship between the clinical efficacy and the underlying molecular mechanisms of demethylating agents remains unclear, especially whether clinical response is a direct result of global demethylation [22,23]. The discrepancy between changes in methylation and clinical effect in several studies suggests that other factors in addition to methylation, such as immune regulation, are involved in patient response. One recent hypothesis is that endogenous retroviruses (ERV), integral parts of the human genome and silenced by methylation, are re-activated upon demethylation by DNMT inhibiting agents. This results in an interferon-like immune response in tumour cells, which finally leads to cell death [23,24]. Whether this indeed explains the therapeutic effect in patients needs to be further investigated. Facilitating this immune recognition and response could therefore be a promising new strategy.

The aim of this study was to examine the effect of decitabine in colon cancer. A translational clinical study was conducted, investigating the effect of pre-operative decitabine on the methylation and expression of WNT target genes *APCDD1*, *AXIN2* and *DKK1* and global methylation in colon cancer patients.

## 2. Materials and Methods

### 2.1. Patient Recruitment and Inclusion Criteria

The DECO study (NCT01882660) was conducted from February 2014 until December 2017. The study was approved by the Medical Ethical Committee of the Academic Medical Center (AMC), Amsterdam. Patients were approached in the outpatient clinic from the AMC and Onze Lieve Vrouwe Gasthuis (OLVG) in Amsterdam and Flevo Ziekenhuis in Almere, all in the Netherlands. Initial clinical staging was performed based on CT scan. Diagnosis was based on endoscopical view, CT scan and/or biopsies, and indication for tumour resection was determined by a multidisciplinary panel. Eligible participants included both male and female patients of 18 years or older, with colon cancer, who had an indication for primary tumour resection. Other inclusion criteria for decitabine treatment included: Karnofsky Performance Score > 70, adequate bone marrow function and adequate hepatic and renal function. Finally, written informed consent had to be signed. Exclusion criteria included known hypersensitivity to decitabine or its additives or if surgery was not planned according to time frame of the study. Moreover, patients who received other systemic or local treatment of the primary tumour in the waiting time until surgery and administration of any experimental drug within 60 days prior to the first dose of decitabine were also excluded.

Pre-treatment samples were taken during endoscopy. For a detailed description of tumour samples, see Section 2.2. Ten ± two days before surgery, patients were treated with decitabine (kindly donated by Janssen-Cilag, The Netherlands) as two one-hour infusions at a dose of 25 mg/m^2^ on two consecutive days. On the day of surgery, directly after resection, a second (post-treatment) sample was taken from the resected primary tumour. Furthermore, pathological staging was performed. Predefined primary endpoint was re-expression of WNT target genes (*APCDD1*, *ASCL2*, *AXIN2* and *DKK1*) measured by quantitative real-time PCR (rt-PCR) in both pre-treatment samples taken during endoscopy and compared with post-treatment samples taken directly after resection. Secondary endpoints included global (*LINE1*) and WNT target gene methylation (*APCDD1*, *ASCL2*, *AXIN2* and *DKK1*) and proliferation assessed by immunohistochemistry in the described pre- and post-treatment tumour samples. During the study, we performed a separate validation study on prediction of prognosis of WNT target gene methylation and showed no additional value of *ASCL2* in analyses [15]. Therefore, results of expression and methylation of *ASCL2* were not included in this study.

To investigate if tumour material from the same patient collected with different procedures (endoscopy vs. resection) was comparable considering methylation, expression and proliferation, a non-treated control cohort was included. Inclusion criteria were patients with colon cancer, older than 18 years, with a Karnofsky performance score > 70. Moreover, if an extra endoscopy procedure was performed, written informed consent was obtained.

### 2.2. Patient Samples

For pre-treatment samples, biopsies from endoscopy were used. Post-treatment samples were collected from resection specimens. In the decitabine-treated cohort, initially, only freshly frozen material was used. In order to obtain fresh-frozen pre-treatment biopsies, an extra endoscopy was performed before surgery. Due to the invasiveness of this procedure, we experienced a low inclusion rate. As a result, the protocol was amended after five patients were included. For the next five patients, FFPE material from a previously performed diagnostic endoscopy was used and compared with FFPE material from surgery.

For the control cohort, six patients with fresh-frozen material were enrolled before the amendment; however, three were excluded. Exclusion reasons were: neo-adjuvant treatment (*n* = 1), only tumour samples from endoscopy were freshly frozen (*n* = 1) and quality of material was insufficient (*n* = 1). We completed the control cohort with twelve colon cancer patients for whom FFPE material was stored. In total, 18 patients were enrolled for the control group, of which 15 could be evaluated for methylation and expression.

For fresh-frozen samples, tissues were immediately snap-frozen using liquid nitrogen and stored at −80 °C. FFPE samples were incubated in 4% buffered formaldehyde for a maximum of 24 h and then transferred to 70% ethanol. Thereafter, samples were dehydrated through 80%, 90%, 96% and 100% of ethanol and finally in 1-butanol and paraffin. For all tumours, multiple (2–5) pre- and post-treatment biopsies were obtained and tumour percentage was determined by HE staining. Two biopsies from each sampling were used for the final analyses. For *LINE1* methylation in treated patients, also a technical replicate was performed and results were averaged for final outcome. For all treated patients, MSI/MSS status, CIMP status and mutation of *TP53*, *KRAS* and *BRAF* were determined. Since the numbers of patients were low, no subgroup analysis could be performed.

Pre- and post-treatment blood samples were collected for haematological toxicity, and Common Terminology Criteria for Adverse Events were used to monitor other toxicities. Pre-treatment blood samples were taken at the time of diagnosis as standard of care and did not require an extra sample. Post-treatment blood samples were taken at the day of surgery.

### 2.3. DNA/RNA Isolation

Genomic DNA (gDNA) and RNA from the fresh-frozen patient samples were extracted with the AllPrep DNA/RNA/miRNA Universal Kit (Qiagen, Hilden, Germany) according to the manufacturer’s instructions. For RNA, RNA integrity number values were determined using the Agilent 2100 bioanalyzer (see Appendix A). FFPE tissue was cut into 10 µm sections and deparaffinised. gDNA was isolated using a Nucleospin DNA FFPE xs kit (Machery-Nagel, Düren, Germany) following the manufacturer’s instructions.

### 2.4. Bisulfite Conversion and Pyrosequencing

Bisulfite conversion was performed with 600–800 ng of gDNA using EZ DNA Methylation-Gold Kit (Zymo research, Irvine, CA, USA) according to the manufacturer’s protocol. For the PCR of the bisulfite converted DNA (bcDNA), PyroMark PCR kit (Qiagen, Hilden, Germany) was used. In short, 20 ng of bcDNA was mixed with kit reagents and a subsequent amplification was performed on a thermocycler. Annealing temperatures were adjusted for different primers: for *LINE1* and *AXIN2*, 56 °C was used; for *APCDD1*, 58 °C; and for *DKK1*, 52 °C. Next, pyrosequencing was performed using 12 ng of bcDNA. PyroMark Assay Design Software 2.0 (Qiagen, Hilden, Germany) was used for primer design and PCR and sequencing primers are listed in Appendix A. For the WNT target genes, the exact location within the gene and CpG sites tested have been described before [15]. LINE1 sequence used was derived from Woloszynska-Read et al. [25]. The sequence analysed was 206–352 (Genbank accession number X52235.1) and contained three CpG sites. For validation, primers were also tested on DNA isolated from FFPE material and compared to DNA obtained from freshly frozen tissue before analysing patient material. This material originated from a previously conducted study, in which xenografts obtained from multiple colon cancer cell lines were used [26]. Results for the validation are shown in Appendix A and show perfect correlation in methylation levels between the two distinct sample preparations.

### 2.5. Quantitative Real-Time PCR

Complementary DNA (cDNA) was synthesised from 1 μg of RNA using Superscript III reverse transcriptase (ThermoFisher, Waltham, MA, USA). For quantitative real-time PCR, 5 ng of cDNA was used in a total reaction volume of 5 µl containing 2.5 µL of SYBR green and 0.5 µM forward and reverse primer (see Appendix A). Reaction was performed in a Lightcycler LC480 II (Roche).

### 2.6. Ki67 Staining

FFPE samples were used for Ki67 staining. Sections of a thickness of 4 µm were prepared and deparaffinised using xylene and rehydrated through ethanol. Antigen retrieval was achieved using 10 mM sodium citrate buffer (pH = 6) (Vector Laboratories, Burlingame, CA, USA) for 20 min at 98 °C. Samples were blocked using Dako REAL Peroxidase-Blocking Solution (Agilent technologies, Santa Clara, CA, USA) for 5 min at room temperature. Ki67 antibody (Sigma, SAB5500134, Saint Louis, MI, USA) was diluted 1:1000, in normal antibody diluent (Klinipath, ABB999, Duiven, The Netherlands), and incubated overnight at 4 °C. After washing with PBS, poly HRP-anti Rabbit IgG (Bright vision, DPVR-55HRP, Immunologic, Duiven, The Netherlands) was added for 30 min at room temperature and finally stained using Bright DAB solution (3,3′ diaminobenzidine, Immunologic, Duiven, The Netherlands). Counterstaining with haemotoxylin (Klinipath, 4085–9002, Duiven, The Netherlands) was incubated for 1 min. After dehydration, slides were mounted using Pertex (HistoLab, Västra Frölunda, Sweden). For material from one patient (patient 9), no staining could be performed due to low quality of the material. For quantification of stainings, haemotoxylin colour was separated from DAB using the plugin “color deconvolution” in ImageJ. Positive nuclei were calculated as a percentage of total nuclei.

### 2.7. CIMP Analysis

For CIMP analyses, a panel of eight genes was used containing *CACNA1G*, *CDKN2A*, *CRABP1*, *IGF2*, *MLH1*, *NEUROG1*, *RUNX3* and *SOCS1*. Furthermore, *ALU* was used to normalise for the amount of input bcDNA. Methods, primers and probes have been described previously [26,27]. Percentage of methylated reference (PMR) > 10 was considered as positive. Tumours where 1–5 out of 8 CIMP markers had a PMR > 10 were defined as CIMP low. Tumours that had ≥ 6 out of 8 markers with a PMR > 10 were defined as CIMP high. Tumours were considered CIMP negative if none of the markers had a PMR > 10.

### 2.8. MSI/MSS

Microsatellite stability was tested during standard of clinical care in the pathology department of our institute using immunohistochemistry for MLH1, MSH2, MSH6 and PMS2. If no results were available, a PCR-based MSI Analysis System, version 1.2 (Promega, Leiden, The Netherlands), was used. In this assay, the markers NR-21, BAT-26, BAT-25, NR-24, MONO-27, Penta C and Penta D were used. Assays were performed according to the manufacturers’ instructions. Samples were considered as microsatellite instable if no staining was present in one of the four immunohistochemical stainings or more than 2 out of 5 markers of the PCR based analyses were instable.

### 2.9. Mutational Status

For mutational status, we used tumour samples obtained from resection. *KRAS* exon 2 and 3 and *TP53* exon 1–11 were amplified by PCR, using 20 ng of gDNA (*KRAS*) or cDNA (*TP53*), 12.5 µL Reddymix (ThermoFisher scientific), 1 µL forward primer and reverse primer 10 µM and 8.5 µL H_2_O in a total volume of 25 µL. For *KRAS*, thermocycler program was as follows: 5 min 95 °C, 40 cycles of 30 s 95 °C, 30 s 50 °C, 1 min 30 s 72 °C, followed by 5 min 72 °C. We used the same protocol for *TP53* only with an annealing temperature of 60 °C. Then, 0.1 µL of the PCR product was sequenced by Big Dye Terminator 1.1 and subsequently analysed by direct Sanger sequencing. Primers are listed in Appendix A. *BRAF* mutation was tested via quantitative rt-PCR with a wild type and *BRAF* V600E specific primer (Appendix A). Reaction was performed using SYBR green by Lightcycler 480. Ct value from *BRAF* mutant was subtracted with Ct value of *BRAF* wild type. Samples with differences of < 4 Ct values were considered as *BRAF* mutant.

### 2.10. Statistical Analyses

The planned maximum sample size for the decitabine treatment group as well as the control cohort was 44 with a 10% loss due to insufficient quality of material. We aimed to include twenty patients with high methylation of WNT target genes and twenty with lowly methylated WNT target genes. The group size was determined based on the incidence of methylation and the expected effect size. We expected a quarter of the tumours to be highly methylated for at least one gene based on results from a previous study [15]. The first interim analysis was performed after ten patients were included and treated with decitabine. For statistical analyses, GraphPad Prism 7 was used. To study the biological effect of decitabine in the clinical samples, we used a paired *t*-test to evaluate significant differences between pre- and post-treatment samples. For comparing the results of the Ki67 staining, a paired *t*-test was used. For all statistical comparisons, the level of significance was set at *p* < 0.05.

## 3. Results

### 3.1. Patient Characteristics

To determine whether decitabine could decrease methylation and thereby re-express WNT target genes, a clinical trial (DECO) was conducted between February 2014 and December 2017. A total of ten colon cancer patients, nine male and one female, were enrolled and pre-operatively treated with decitabine, after which we performed an interim analysis that is reported here. Baseline characteristics and flow chart for inclusion are presented in Figure 1.

The median age was 65 years (range 59–78) and all patients were evaluable for toxicity, and, from all patients, material was available for methylation analyses. Furthermore, a non-decitabine-treated control cohort (*n* = 18) was enrolled, of which 15 patients were eligible for analyses. Although clinical outcome was not an endpoint of this study, three-year overall survival was documented and all patients reached this endpoint. In the follow-up after four years, two events occurred.

### 3.2. Effect of Pre-Operative Treatment with Decitabine on Methylation in Colon Cancer Patients

Before performing analyses on our primary endpoint, we verified if biopsies had a comparable percentage of methylation to tumour samples from resection using a control cohort. In this cohort, a total of 18 patients were included, of which 15 patients were available for analyses. To determine levels of global methylation, *LINE1* methylation was used as a surrogate marker [28]. In this cohort, the average of *LINE1* methylation for biopsies was 69.0 ± 6.1% and resection material was 69.0 ± 4.3% (*n =* 15). Paired analysis revealed that biopsies and resection material could be directly compared, showing no statistical differences in methylation (*p =* 0.9718) (Figure 2A). Next, *LINE1* methylation was assessed in decitabine-treated patients (*n =* 10). The average *LINE1* methylation from the ten patients before treatment, as analysed on the biopsy material, was 71.2 ± 6.4%, while after treatment, the average was numerically lower (67.2 ± 6.5%). Paired analysis of the patients indicated that all but one patient showed a decrease in methylation and that this was significant when analysing the group (*p =* 0.0075) (Figure 2A). Nevertheless, this decrease was relatively small for all patients tested, indicating that decitabine could modulate *LINE1* methylation, but with the dosing used, the impact was minimal. To determine the impact of decitabine on WNT target gene methylation, CpG methylation of *APCDD1*, *AXIN2* and *DKK1* was measured in the first five patients from which fresh-frozen samples were available. Importantly, analysis of five patient sample pairs showed similar WNT target CpG methylation in pre- and post-treatment samples (Figure 2B) and no clear decrease could be detected. However, firm conclusions cannot be drawn for these data due to low patient numbers and the fact that these patients did not display high WNT target methylation at the start of treatment.

### 3.3. Effect of Pre-Operative Treatment with Decitabine on Gene Expression and Proliferation in Colon Cancer Patients

Despite the fact that only a small difference in LINE1 methylation and no clear impact on WNT target methylation could be detected, differences in gene expression or cell biological features, such as cell proliferation, could potentially be orchestrated without overt changes in methylation. To this end, WNT target gene and LINE1 expression was first analysed with quantitative rt-PCR in the fresh-frozen samples (*n* = 5). This revealed that both LINE1 and WNT target gene expression were not significantly different between pre- and post-treatment tumour samples (Figure 2C).

A reduction in proliferation after treatment with decitabine has been reported [29] and could also significantly impact the efficacy of demethylation as this is suggested to require cell cycle progression. Therefore, immunohistochemical staining for Ki67 was used to determine the expression on protein level (Figure 3). A large variation in Ki67 positive cells between patients as well as tumour region was observed, which is in line with earlier results [30]. However, no consistent difference between pre-treatment samples and post-treatment samples was detected (Figure 3B) (*p =* 0.7618). Although this may relate to the relatively small group size, we conclude that a strong impact on proliferation was not detected. This likely aligns with the lack of impact of a short course decitabine treatment on tumour growth.

### 3.4. Decitabine Does Not Induce Expression of Endogenous Retrovirus ERVL and Interferon Associated Genes in Colon Cancer Patients

Endogenous retroviruses (ERVs) are heavily encoded in our human genome and effectively silenced by CpG methylation. Recent evidence suggests that decitabine can result in effective demethylation of these silencing CpG islands and result in reactivation of ERVs [23,24]. The cellular response towards ERV reactivation is rapid induction of interferon and interferon-related gene expression mounting an anti-viral and, as a result, anti-tumour immune response [23]. Importantly, as recent studies have also suggested that the inhibition of immune checkpoints in a neo-adjuvant setting is effective in colorectal cancer [31], we wondered whether decitabine could activate ERVs and hence provide an anti-tumour response. Therefore, the impact of decitabine on the gene expression of interferon-related genes and ERV *ERVL* was assessed in our patient tumour samples (*n* = 5; DECO patient 1–5). However, although the patient numbers were limited, neither the reactivation of *ERVL* nor the activation of the interferon response was evident (Figure 4), suggesting that the levels of decitabine used in this study do not lead to ERV activation.

### 3.5. Toxicity

Grade 1 adverse events are summarised in Appendix A and pre- and post-treatment laboratory tests of the patients in Appendix A. In addition, decitabine administration had no effect on the timing of surgery. This indicates that decitabine can be used safely pre-operatively at the concentrations and timing used in this study.

### 3.6. Study Closure

Due to low patient inclusion, we amended the protocol after two years to use FFPE material to avoid the need for an extra endoscopy for patients. After including ten patients (not pre-specified) for the decitabine arm, the current analysis was performed. This revealed that decitabine treatment with the employed scheme did not result in demethylation and/or subsequent upregulation of WNT target genes partially controlled by methylation. The effect of decitabine on *LINE1* methylation was significant, yet too limited to be impactful when analysing the expression of *LINE1*, while we also did not observe an impact on WNT target methylation or expression or on *ERV* expression. Initially, we anticipated to include forty evaluable patients for decitabine treatment, aiming to change WNT target expression and methylation. However, with the results of the first ten patients, a different conclusion after forty patients was unlikely and we closed the study for further patient inclusion to avoid unnecessary impacts on patients.

## 4. Discussion

For early-stage colon cancer, surgery remains the cornerstone of treatment. However, in the case of stage III or high-risk stage II disease, adjuvant treatment with cytotoxic drugs improves patient outcomes. Although an overall survival benefit for cytotoxic treatment in these groups has been clearly documented, the proportion of patients with increased survival because of adjuvant therapy remains low. In clinical stage I-III colon cancer, neo-adjuvant therapy is currently not standard therapy, but a recent study shows that neo-adjuvant FOLFOX is safe, with no increase in perioperative morbidity [32]. This not only paves the way for studies assessing the long-term benefit of this strategy but also for the use of neo-adjuvant treatment for expeditious evaluation of response and thus for the development of new therapeutic options in colon cancer. CpG hypermethylation is an important event in tumourigenesis and its reversible nature makes it an attractive target for therapy. In addition to its role in tumour progression, CpG hypermethylation is associated with poor prognosis [14,33,34], underlining the relevance of studying the effect of demethylating agents in colon cancer.

In this study, the biological effect of the demethylating drug, decitabine, was studied in colon cancer patients. A translational clinical study was conducted in which patients were treated with decitabine prior to surgery. The impact on *LINE1* methylation using this treatment was significant but, compared to earlier studies, very small. In agreement, the observed decrease did not lead to an increase in *LINE1* gene expression, nor did we observe a change in methylation of WNT target genes or WNT target gene or ERV expression. The minor effect on methylation was further corroborated by the observation that tumour proliferation did not change in the treated patients. In conclusion, this suggests that dosing decitabine twice at the concentration used, and after 8–12 days, is not sufficient to obtain impactful changes in tumour cells. It should be mentioned that the study was, due to pre-mature closure, underpowered. Furthermore, none of the patients analysed for WNT target gene methylation showed high levels of methylation for these genes and this could hamper the effect of decitabine. However, this level of methylation was not unexpected, as a previous study showed positive WNT target gene methylation in 26% of the tumours [15]. Nevertheless, the low methylation levels were also not affected by decitabine treatment, nor was the expression of the WNT target genes. In line with the limited effect on demethylation, a re-activation of endogenous retrovirus *ERVL* as well as the linked interferon response was not observed in tumour material from a limited number of colon cancer patients treated with decitabine.

There are several explanations for the limited effect of decitabine in the patients in our study. Firstly, to prevent toxicity and, more importantly, an impact on the timing or success of the surgery, a relatively low dose of decitabine at only two injections was used, which could have resulted in a relatively low effective concentration in the patients. However, data from previous clinical studies suggest that the dose and timing that was used could be appropriate for demethylation in patients with solid tumours or myelodysplastic syndrome [35,36]. Nevertheless, repetitive administration and longer time intervals have been reported to optimise the effect [35,37,38,39]. Another explanation for our findings could be that decitabine is less effective in tumour tissue than in PBMCs, which are commonly used to monitor the effect on DNA methylation in patients [35,40,41]. Studies that compared the effect of methylation in PBMCs with tumour samples are limited and show conflicting results [35,42,43,44,45]. It is therefore difficult to extrapolate results from PBMCs to tumour samples as PBMCs are, by virtue of their location, more accessible for decitabine. PBMCs were not collected during our study as our focus was on the resected tumour tissue. However, for additional information about treatment schedules, collection of PBMCs might be useful in future studies. Moreover, collection of circulating tumour DNA could be insightful to evaluate response on methylation and prevent invasive biopsies [42,45,46,47]. Finally, a lower proliferation rate in tumour cells in patients compared to xenografts or in vitro cultures could also impair the effect of decitabine. Decitabine is only active during cell proliferation and demethylation is progressive with each cell division. Although the proliferation rate in these patient samples measured by Ki67 staining was relatively high, this still could be lower than in vitro. Nevertheless, no correlation between Ki67 staining in pre-treatment material and effect on *LINE* methylation was observed (data not shown).

The findings of a demethylating agent in colon cancer patients in this study are in line with several previously reported clinical studies in solid tumours [48]. Thus far, four studies on demethylating agents in colon cancer have been conducted. In a clinical trial using a combination of decitabine with panitumumab, a 10% response rate was shown, but no effect on *MAGE* re-expression was observed [9]. In another study, capecitabine and oxaliplatin were combined with azacitidine in twenty-six colon cancer patients [46]. In this study, no objective response was observed, neither in CIMP high nor in CIMP low patients. In 60% of patients, methylation of vimentin was decreased; however, this effect was limited and did not outperform technical variation of methylation testing. More recently, guadecitabine was combined with irinotecan to treat metastatic colorectal cancer patients [37]. No consistent *LINE1* demethylation was detected in tumour biopsies or circulating tumour DNA after 8 days. However, a reduction was seen after 15 days, although no correlation with clinical response could be observed [37]. This was in line with results from a study with 47 colon cancer patients, where demethylation was shown in post-treatment samples but was unrelated to response or overall survival [49]. Overall, the results of trials with demethylating agents in colon cancer are disappointing, although responses in individual patients are seen, emphasizing the importance of biomarkers to predict response or to find synergy with other drugs, e.g., immune checkpoint inhibitors.

Despite our findings, the study setup in which patients received neo-adjuvant treatment for a short period and tissue pre- and post-treatment was analysed is of interest for future drug studies. Our method facilitated the measurement of treatment response in colon cancer patients on primary tumour tissue on individual basis in a short time frame. Recent data on neo-adjuvant immunotherapy in colorectal cancer patients support this approach, and a more extensive analysis of the role of neo-adjuvant therapy in colon cancer is warranted. This study setup also allows for a quick evaluation of hypotheses and drugs that emanate from preclinical work. If an effect is detected, this strategy would allow for a rapid dissemination and identification of biomarkers to select patients for certain treatments. Thereby, this setup could potentially be used to personalise adjuvant treatment in colon cancer.

## 5. Conclusions

No decrease in WNT target gene methylation was observed after short-term pre-operative treatment with decitabine in a limited amount of tumour tissue from five colon cancer patients. Future methylation studies should focus on optimisation of treatment regimens in patients with highly methylated tumours and perform parallel collection of PBMCs with tumour material.

## Figures and Tables

**Figure 1 cancers-13-02357-f001:**
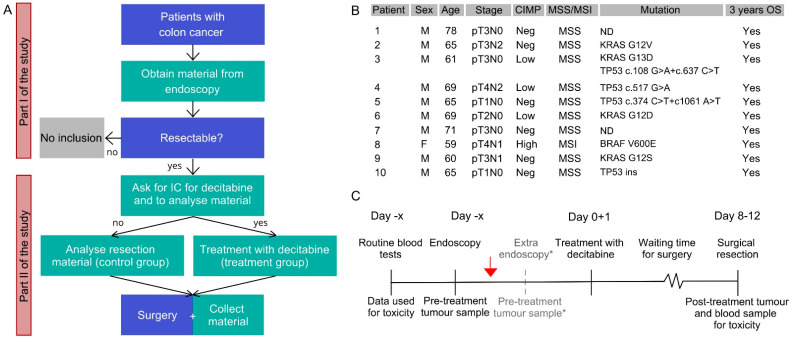
(**A**) Flow chart of DECO study. In blue squares, daily routine steps in clinical care are shown; in green squares, additional steps for DECO study are shown; (**B**) Baseline characteristics of decitabine-treated patients included in DECO study. Indicated stage is pathological staging after resection was done. Initial staging (for inclusion) was performed based on CT scan; (**C**) Timeline for DECO study. Red arrow shows the moment of inclusion. Blood tests before inclusion were part of standard of care. * For the patients with fresh frozen material, an extra endoscopy was performed to obtain freshly frozen biopsies as pre-treatment sample. For FFPE patients, pre-treatment samples were obtained from the diagnostic endoscopy performed for clinical purposes. IC = informed consent, ND = not detected, OS = overall survival.

**Figure 2 cancers-13-02357-f002:**
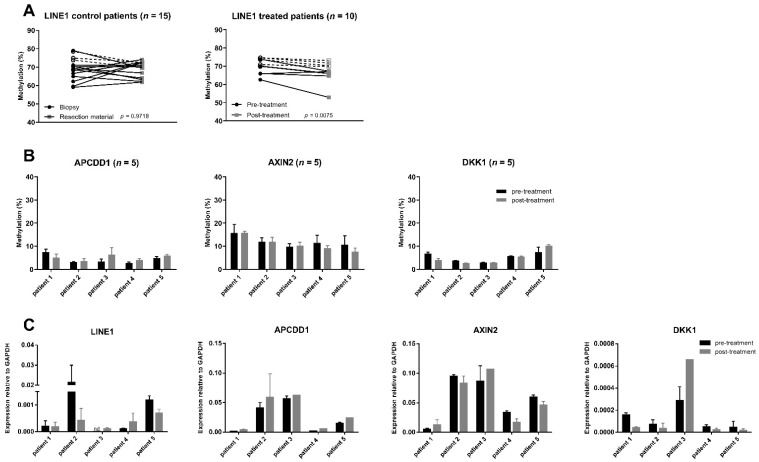
(**A**) Methylation of *LINE1* in the control group (*n* = 15) and in the treated group (*n* = 10) (before and after treatment with 25 mg/m^2^ decitabine two times) measured by pyrosequencing. In both cohorts, FF tumour samples and FFPE samples were included. Open symbols and dotted lines represent FF samples, and closed symbols and lines represent FFPE samples. In the treated cohort, two technical replicates per time point were averaged and two biological replicates (two different samples from the same tumour) were used. For patient 6 to patient 10, no biological replicate was available for the pre-treatment sample. For statistical analyses, for pre-treatment and post-treatment samples, the average of all measurements was used. A paired *t*-test revealed no significant difference in the control cohort (*p* = 0.9718). For the treated cohort, a significant (*p* = 0.0075) difference was shown; (**B**) Methylation of WNT target genes before and after treatment with decitabine in colon cancer patients measured by pyrosequencing (*n* = 5); (**C**) Expression of *LINE1* and WNT target genes after treatment with decitabine measured by quantitative rt-PCR in fresh-frozen samples (*n* = 5). Values are the average of two samples (both for pre- and post-treatment samples), except for patient 3, where only one post-treatment sample was available.

**Figure 3 cancers-13-02357-f003:**
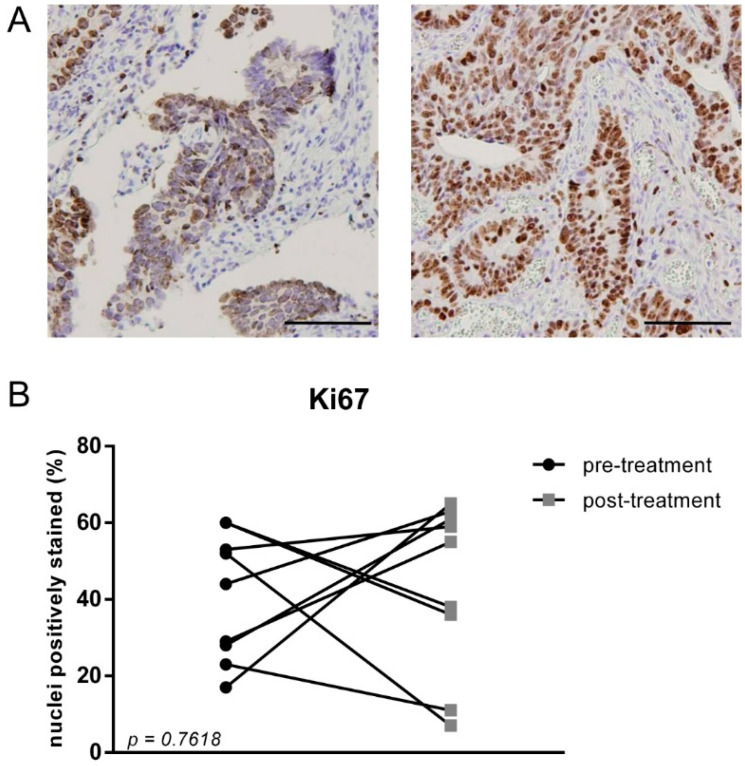
(**A**) Representable Ki67 staining of biopsy and resection material of the tumour from one patient. The scale bar represents 100 µm; (**B**) Percentage of Ki67 positive cells compared to total cells from nine treated patients. A representative area of the tumour block was used for quantification. No significant difference between pre- and post-treatment samples (*p* = 0.7618) was observed using a paired *t*-test.

**Figure 4 cancers-13-02357-f004:**
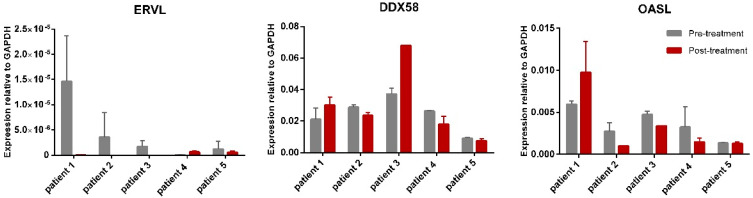
Expression of *ERVL* and interferon genes *DDX58* and *OASL* in pre- and post-treatment samples (*n =* 2 per patients except from patient 3) from patients treated with decitabine. Only fresh-frozen samples were used.

## Data Availability

No new data were created or analyzed in this study. Data sharing is not applicable to this article.

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
