# Peer review of "Pre-Operative Decitabine in Colon Cancer Patients: Analyses on WNT Target Methylation and Expression"

_cancers, 2021, doi:10.3390/cancers13102357_

Round 1

Reviewer 1 Report

In this paper, Linnekamp and colleagues reported the results of the trial NCT01882660. Primary endpoint of this trial was to assess whether short-course pre-operative treatment with decitabine increases Wnt target gene expression as measured in resected tumors compared to pretreatment biopsies, in patients with primary colon cancer. The study was closed to insufficient accrual and the paper reports the results from a cohort of 10 patients treated with decitabine. Although the premises are interesting, there are major issues in the study.

Despite the interesting setting in which this study is performed (neoadjuvant), several limitations of this study were found. First, it was not clear how disease staging was performed (CT scan?) and if treatment outcome in patients were annotated. In addition, patients should have been selected based on methylation status while all patients were included without considering baseline tumor methylation. Indeed, if decitabine would have been given to patients with hypermethylated colorectal cancer, it might have led to greater impact in terms of demethylation.

The most important is that statistical section do not provide information about planned sample size and design. Since the original planned sample size was 44 patients, the trial is clearly underpowered to draw meaningful conclusions, i.e. both type I and II errors are unknown but clearly very large, so that also the conclusion that decitabine do not increase WNT target gene expression is questionable.

However, here following I enlisted some comments aiming to improve the manuscript submitted by the authors.

Comments:

  • The introduction section is too long and it should be shortened. Furthermore, I suggest starting with a brief description on the role of methylation in all tumors than focusing on colorectal cancer.
  • Terms identifying genes should be written in Italic. Please modify accordingly in the manuscript.
  • Page 4 line 128. What do the author mean with “high suspicion of colon cancer”? If a biopsy of primary tumor was performed why should they state “suspicion”? How was tumor staging performed? Please improve methods section at this regard.
  • Page 4 line 131. The “exclusion criteria” statement is too long. In suggest splitting it in two parts.
  • In this study patients with resectable colorectal cancer ranging from stage I to III high risk were included. However, the authors demonstrated the role of WNT genes methylation in stage II disease. Are there any data also in stage III disease?
  • Figure 1 is grainy and therefore it should be improved in terms of resolution.
  • Why did not the authors include patients with hypermethylated CpG island in this study?
  • Was there any response or tumor shrinkage to treatment with decitabine in this subset of patients?
  • As the Introduction, also the discussion section is too long and need to be shortened.

Reviewer 2 Report

Linnekamp et al. investigated the effect of decitabine on DNA methylation profile in colon cancer patients. Although the clinical study has a limited number of patients, this is not the main limitation. One can accept that a homogenized study would not benefit from a high number of individuals. However, the whole manuscript is messy, not focused, with many mistakes and unclear. I recommend either rewrite the manuscript or reject it.

The whole paragraph about ERV is confusing and not focused. It is not clear how the authors came to that hypothesis, and they want to prove it. And the connection with neoadjuvant immunotherapy, decitabine and DNA methylation status miss logic. I rather like the clarification in the result section than in the introduction.

Section 2.1 is not clear, messy, and not focused. I strongly advise rewriting this section. My points are below:

Although no young patients were included in the study, the inclusion criteria set at 18 years (and older) is rather critical. It is known that around this age the familiar syndrome manifests and cancer of sporadic origin is under representative. Besides, patients diagnosed at lower age evinced worse prognosis and represent a defined group of patients.

Row 128 What does it mean “high suspicion of colon cancer”? Patients included in the study had histologically proved incident colon cancer or not?

Write in detail, how many individuals were included in the study.

Row 147 Authors stated that the second sample was taken from the tumor. However, there is completely missing information about the first sample.

The concrete time (days) of Pre-and post-treatment blood samples is missing. I mean when exactly how many days before decitabine administration was blood collected, as well as how many days after the end of the therapy. The authors stated that blood was collected for toxicity assessment. In the results section, this part is almost omitted.

Rows 148-155 are confusing; it is not clear what was measured and why.

For comparison of DNA methylation isolated from fresh frozen and FFPE material, first, the confirmation should be performed within at least one patient to determine the extent of their similarity. The FFPE material is tricky and methylation at many CG sites might be lost.

Row 204 validation of what to what.

Rows 206-208 It is not clear which material and what was the aim.

Row 211 Complementary DNA!!!!

The statement of quality RNA is missing. What RIN number did you obtained?

Row 240 Tumours where ≥ 6 or 1-5 out of 8 CIMP markers had a PMR > 10, 240were defined as CIMP high or CIMP low, respectively.    It is not clear at all.

For MSI status, which markers were used?

Row 254 it is not clear which material was used for the assessment of TP53 mutation.

Row 260 I am not sure whether the methodology for BRAF mutations assessment is appropriate.

Figure 1, please state in the table which particular TP53 mutations were identified and used term for gene not for protein.

Authors: Before performing analyses on our primary endpoint, we verified if biopsies had percentage of methylation as tumour samples from resection using a control cohort. To assess the effect of decitabine on global methylation, LINE1 methylation was used as a surrogate marker.    I thought that the control group did not get decitabine and from these sentences, it seems that they did get it. Or the authors wanted to say that they wanted to see how the methylation profile is changing through the analysis to observe the baseline changes in DNA methylation profiles that will be further compared to decitabine treated patients? Please make this point clear.

Many patients were collected three years ago, or even before a longer period thus author might have the chance to collect the clinical outcomes. I would recommend comparing the decitabine group to non-treated and observed whether there is any impact on prognosis or disease recurrence. The study would benefit from this outcome.

Minor:

Row 92  MDS abbreviation is not defined

Row 97 was observed.

Row 112, 115 venue?

Reviewer 3 Report

Dear Prof. Dr. Medema, 

thank you for that interesting study analysing the effect of the demethylating agent decitabine on methylation and expression of negative regulator of  Wnt signaling as well as expression of endogenous retroviral genes having an immune stimulatory  effect. 

Although pre-operative short term treatment with decitabine neither decreases WNT target gene methylation nor has an effect on the expression of endogenous retroviral genes, the results are of great importance for future study design. 

In your patient characteristics, mutational profile of KRAS, BRAF and P53 was determined, but not APC/CTNNB1. Why do you think it is not important to include the mutational status of APC/CTNNB1? Sato et al showed in 2007 that DKKs could not inhibit b-catenin/TCF reporter activity in CRC cells with constitutive Wnt signaling due to APC mutations. Is there a mechanism known how constitutive Wnt signalng affect Wnt target gene methylation of negative Wnt regulator DKK1, AXIN2 ? 

Thank you very much. 

Best regards

Reviewer 4 Report

The manuscript by Linnekamp et al. entitled "Pre-operative decitabine in colon cancer patients analyses on WNT target methylation and expression" explores whether treatment with the demethylating agent decitabine could potentially alter WNT pathway genes and reactivate retrotransposons leading to genomic instability in stage II colon cancer. The hypothesis is based on murine models showing some promise for this DNMT inhibitor therapy approved for several hematological cancers. The results of a negative finding trial that was terminated early are important to report for reference in the literature, as well as the lessons learned with recruitment are key for researchers in the future.

Comments:

Lines 157-159: This statement is confusing. What do you mean here? Did those 18 patients have endoscopy and then resection? Please clarify.

Lines 169-171: How different were the results between the fresh frozen and the FFPE. How degraded was the DNA after restoration protocols? Do you have a way to quantify these differences?

Lines 196-208, Table S1: Could you specify which CpGs were measured in the sequence and why? I am particularly confused with LINE1. Where is your sequence located? Is this L1Hs? please refer to the GENEBANK sequence and the specific nucleotide(s) measured? If the average of several CpGs, specify which ones and how different were the measurement among them. Please confirm all the sequences used, state the CpGs measured, the variability and the CV  of your pyrosequencing. If you used a different primer to the most classically reported, could you state why?

Results lines 294-297 and Figure 2 A. Please specify the measured levels of FFPE and fresh frozen samples for biopsy and treatment.

Figure 2B, results: add a note in the caption specifying that the n for WNT measurements was only 5.

Table S4: It was not possible to evaluate this table from Zenodo. There was an error while downloading, please check.

Round 2

Reviewer 2 Report

I have no other comments.

Author Response

We thank this reviewer for the original review and the positive comments

Reviewer 4 Report

The manuscript by Linnekamp et al. entitled "Pre-operative decitabine in colon cancer patients analyses on WNT target methylation and expression" explores whether treatment with the demethylating agent decitabine could potentially alter WNT pathway genes and reactivate retrotransposons leading to genomic instability in stage II colon cancer. The hypothesis is based on murine models showing some promise for this DNMT inhibitor therapy approved for several hematological cancers. The results of a negative finding trial that was terminated early are important to report for reference in the literature, as well as the lessons learned with recruitment are key for researchers in the future.

The authors have addressed all the comments

Author Response

We thank this reviewer